# Novel Microsatellite Tags Hold Promise for Illuminating the Lost Years in Four Sea Turtle Species

**DOI:** 10.3390/ani14060903

**Published:** 2024-03-14

**Authors:** Tony Candela, Jeanette Wyneken, Peter Leijen, Philippe Gaspar, Frederic Vandeperre, Terry Norton, Walter Mustin, Julien Temple-Boyer, Emily Turla, Nicole Barbour, Sean Williamson, Rui Guedes, Gonçalo Graça, Ivan Beltran, Joana Batalha, Andrea Herguedas, Davide Zailo, Vandanaa Baboolal, Francesca Casella, George L. Shillinger

**Affiliations:** 1Upwell, Monterey, CA 93940, USA; nicole.ann.barbour@gmail.com (N.B.); sean.alexander.williamson@gmail.com (S.W.); 2Mercator Ocean International, 31400 Toulouse, France; pgaspar@mercator-ocean.fr (P.G.); jtempleboyer@mercator-ocean.fr (J.T.-B.); 3Aquarium La Rochelle, Centre d’Etudes et de Soins pour les Tortues Marines, 17000 La Rochelle, France; 4FAU Marine Science Laboratory, Department of Biological Sciences, Florida Atlantic University, Boca Raton, FL 33432, USA; jwyneken@fau.edu (J.W.); eturla2013@fau.edu (E.T.); 5Lotek Wireless, Inc., Havelock North 4130, New Zealand; pleijen@lotek.com; 6Institute of Marine Science, IICM Okeanos, University of the Azores, 9901-862 Horta, Portugal; frederic.vandeperre@gmail.com (F.V.); goncalo@flyingsharks.eu (G.G.); m.joana.batalha@gmail.com (J.B.); a.herguedas@outlook.es (A.H.); 7Institute of Marine Research, IMAR, 9900-138 Horta, Portugal; 8Georgia Sea Turtle Center, Jekyll Island Authority, Jekyll Island, GA 31527, USA; terrymhnorton@gmail.com (T.N.); dzailo@jekyllisland.com (D.Z.); 9Cayman Turtle Conservation and Education Center, West Bay, P.O. Box 812, Grand Cayman KY1, West Bay 1303, Cayman Islands; wgmustin@turtle.ky (W.M.); vandanaababoolal@turtle.ky (V.B.); francesca.casella3@gmail.com (F.C.); 10Department of Environmental Biology, SUNY College of Environmental Science and Forestry, Syracuse, NY 13210, USA; 11School of Biological Sciences, Monash University, Clayton, VIC 3800, Australia; 12Flying Sharks, 9900-124 Horta, Portugal; guedes@flyingsharks.eu (R.G.); ivan@flyingsharks.eu (I.B.); 13MigraMar, Bodega, CA 94923, USA

**Keywords:** microsatellite tag, performance analysis, satellite tracking, early juvenile sea turtle, lost years, diving behavior, North Atlantic Ocean

## Abstract

**Simple Summary:**

Observing juvenile sea turtles at sea is challenging due to their small sizes and cryptic behaviors and is compounded by the vastness of the ocean. The first phases of sea turtle life history, commonly known as the “Lost Years”, remain enigmatic and poorly understood, despite significant advances in animal ad ocean observation technologies. Our study focused on testing new prototypes of microsatellite tags and analyzing their performance on 160 juvenile sea turtles of four species in the North Atlantic. The results demonstrated that, despite challenges brought by miniaturization, the tags proved effective. However, the tracking periods were shorter than expected, limiting our ability to deeply study and understand the turtles’ dispersal features. We found that tracking durations varied among species, indicating some limitations due to certain behaviors such as neritic interactions or diving activity by young sea turtles, and revealing that some of their behaviors may be too strenuous for small tags. Our findings have important implications for the bio-logging community, especially those studying marine animals such as sea turtles. Our study showcases recent technological advances and contributes to improving the effectiveness and durability of miniaturized satellite tags deployed within the marine environment. Our methodologies and findings have improved our understanding of the “Lost Years”, promise to inform ongoing future technological advances and contribute to increasing the effectiveness of conservation efforts.

**Abstract:**

After hatching, sea turtles leave the nest and disperse into the ocean. Many years later, they return to their natal coastlines. The period between their leaving and their returning to natal areas, known as the “Lost Years”, is poorly understood. Satellite tracking studies aimed at studying the “Lost Years” are challenging due to the small size and prolonged dispersal phases of young individuals. Here, we summarize preliminary findings about the performance of prototype microsatellite tags deployed over a three-year period on 160 neonate to small juvenile sea turtles from four species released in the North Atlantic Ocean. We provide an overview of results analyzing tag performance with metrics to investigate transmission characteristics and causes of tag failure. Our results reveal that, despite certain unfavorable transmission features, overall tag performance was satisfactory. However, most track durations were shorter than those observed on individuals of similar size in other studies and did not allow for detailed analyses of trajectories and turtle behavior. Our study further suggests that tracking durations are correlated with the targeted species, highlighting a lack of robustness against some neritic behaviors. Unprecedented diving data obtained for neonate sea turtles in this study suggest that the vertical behaviors of early juveniles are already too strenuous for these miniaturized tags. Our findings will help to inform the biologging research community, showcasing recent technological advances for the species and life stages within our study.

## 1. Introduction

The emergence of satellite tracking technologies has revolutionized the study of highly migratory marine species, including birds, mammals, fish, and reptiles such as sea turtles [1]. These advancements include the ability to track animals over long distances and extended periods of time without the need for physical recovery of the tracking device and by directly relaying obtained data via satellites. While early tracking studies using Platform Transmitters Terminals (PTT) were relatively short in duration [2,3], data from recent studies have demonstrated that tracking durations may exceed one year [4]. Today, these technologies have become a fundamental tool for studying the movement and spatial ecology of marine species [5], with thousands of devices deployed each year [6]. They allow for relatively accurate long-term tracking of marine animal migrations and can provide insights into a variety of other knowledge gaps, including habitat use at the scale of entire oceans [7], navigation mechanisms [8,9,10], and faced threats and risks [11].

While satellite technology is widely used for tracking adult and subadult sea turtles [12,13], it remains challenging as an application for very young individuals due to their small size and weight [13,14,15]. This challenge has historically resulted in a critical lack of observational data on the dispersal of neonate and small juvenile sea turtles from the time they leave their nesting beaches until they are encountered many years later in their foraging areas or near their natal beaches. These early years remain highly enigmatic for many sea turtle species and are often referred to as the “Lost Years” [16]. During this developmental period, young individuals disperse across entire oceans through various ocean currents, encounter different ecosystems, and face myriad threats [14,17,18,19] that highly impact their survival at the population level [20]. In order to implement effective conservation measures to protect entire sea turtle populations, the “Lost Years” life stage must be prioritized for observation and increased understanding.

Monitoring sea turtles during early developmental stages requires the use of appropriately sized (e.g., tag and attachment weighing <5% body mass, limited drag to minimize hydrodynamic cost) [13,21,22,23] electronic devices. Satellite tag miniaturization efforts have proven challenging, and manufacturers have encountered a variety of technical obstacles in their efforts to make tags smaller and lighter [24]. Size reduction requires the integration of diverse functionalities into a compact package, often involving limitations in terms of power supply, data storage, and transmission capabilities [24]. Miniaturization also poses challenges in terms of durability and robustness, as these small devices must withstand the harsh environmental conditions inherent to the marine environment and the movements of the individuals carrying them [4,13,14,25,26].

Challenges aside, tracking the oceanic movements and behaviors of early-stage sea turtles is a burgeoning field fueled by ongoing advancements. Satellite tag miniaturization has facilitated successful deployment of tracking devices on very young juvenile loggerhead (*Caretta caretta*) and green turtles (*Chelonia mydas*) as small as 15 cm straight carapace length (SCL) and juvenile Kemp’s ridleys (*Lepidochelys kempii*) of about 30 cm SCL, with some tracking durations spanning more than 2 months [18,19,27,28,29]. However, only a few telemetry studies of very small (<10 cm) neonate juvenile leatherbacks (*Dermochelys coriacea*) have been reported, all using acoustic technologies and resulting in very short tracking durations (<2 h) and relatively nearshore trajectories [30,31,32]. No satellite-based telemetry studies of equivalently sized juvenile leatherbacks exist within the literature.

We analyzed the performance of 164 innovative Argos-based microsatellite tags (≈2–5 g) and examined their transmission features. To test performance, 160 turtles and four passive drifters were fitted with microsatellite tags. We investigated their tracking durations and probable causes of failure, considering the influence of the type of technology used and the behavior of the specific sea turtle species equipped with the tags. Our research summarized three years of testing and significantly improved our understanding of the challenges and limitations associated with tracking early juvenile sea turtles. Our findings will inform efforts to improve the effectiveness and durability of miniaturized satellite tags deployed within the marine environment. These results contribute to the improvement and development of future tracking technologies and methodologies and are an important step forward in unveiling the dispersal of juvenile sea turtles.

## 2. Materials and Methods

### 2.1. Overview of the Study

We equipped juveniles of four species of sea turtles (green [CM], leatherback [DC], loggerhead [CC], and Kemp’s ridley [LK]) in the North Atlantic Ocean with prototypes of microsatellite PTTs (hereafter mentioned as tags) developed by Lotek Wireless, Inc. (Havelock North, New Zealand). All individuals were either reared in captivity, wild-caught, or rehabilitated. Turtles from the Azores (Portugal) were either wild-caught or retrieved as stranded and temporarily housed and/or rehabilitated. The turtles were subsequently released with the help of local organizations from four locations: (Figure 1) Florida (USA), The Cayman Islands, Jekyll Island (GA, USA) and the Azores. Upon release, we tracked the horizontal and vertical (for individuals bearing tags equipped with a pressure sensor) movements of early-stage sea turtles, with the goal of providing valuable information on their behavior and environmental interactions (e.g., use of eddies [33,34], thermoregulatory diving activity [35,36]) during this enigmatic life stage.

### 2.2. Capture and Husbandry

#### 2.2.1. Florida

All juvenile turtles from Florida (leatherbacks, loggerheads, and green turtles) were initially collected as part of an unrelated study led by researchers at Florida Atlantic University (FAU), to examine primary sex ratios. The turtles were collected upon natural emergence from in situ nests laid along the coast of Boca Raton or Juno Beach (FL, USA), located on Florida’s Southeastern coast. All captive-reared turtles were maintained in natural seawater, had a 12 h/12 h light/dark cycle, and were fed an in-house manufactured gel diet specific to species until they reached 100+ grams. All these turtles were in the laboratory for approximately 3–4 months for laparoscopic sex identification; they were fully recovered and in good health and body condition when equipped with microsatellite tags and released.

#### 2.2.2. The Cayman Islands

Juvenile green turtles released around the Cayman Islands were obtained from the Cayman Turtle Conservation and Education Centre (CTCEC). The turtles were hatched and raised in a captive setting, reared in various-sized concrete tanks supplied with direct unfiltered seawater, fed a nutritionally complete, floating, and modified extruded fish diet, and kept in groups of like size and age until they reached the desired age class for release [37]. All turtles were quarantined (>6 months) prior to release and were checked to ensure that they each fulfilled requisite health and quarantine requirements for release [37]. Thus, all the released turtles were designated to be in good health and body condition before the deployment of tags and the release [37].

#### 2.2.3. Azores

All juvenile loggerheads from the Azores were wild-caught, except for one individual found stranded in Porto Pim beach, Faial. The wild-caught individuals were captured by hand or with a dip net in the vicinity of the islands of Faial and Pico. One turtle was missing the left hind foot, but the wound was well healed. All wild-caught turtles were maintained in natural seawater in the Porto Pim Aquarium, operated by Flying Shark, Lda., and subject to a natural light:dark cycle. The turtles were maintained for a period ranging from 8 to 166 days (median = 37 days) prior to release. The one stranded turtle was maintained in the same circumstances for a period of 213 days prior to release. All individuals were in good condition at the time of release.

#### 2.2.4. Jekyll Island

Juvenile Kemp’s ridley turtles released off Jekyll Island (GA, USA) were collected while stranded and cold-stunned (hypothermia) off the coast of Cape Cod (MA, USA). All these turtles were diagnosed with bacterial pneumonia and were rehabilitated at the New England Aquarium and other facilities in Massachusetts (USA). Upon recovery the turtles were transported to the Georgia Sea Turtle Center (GSTC) on Jekyll Island, where they received extensive diagnostics, and, if necessary, additional treatments until the waters warmed and they were ready to be released. One juvenile loggerhead was recovered from a nest on Jekyll Island that had been invaded by fire ants. This turtle was rehabilitated at the GSTC until its release almost 15 months later. All individuals were in good condition at the time of release.

### 2.3. Turtles’ Sizes and Growth Curves

We recorded carapace length for each turtle prior to release (Table A1, Table A2, Table A3 and Table A4, Appendix A). The Straight Carapace Length (SCL) and/or Curved Carapace Length (CCL) were measured using several standard methods (notch-to-notch, notch-to-tip, averages between both) depending on the release site. Within this study, measurements were combined into a single observational dataset, independently of the measurement technique. In cases where the SCL was missing, it was computed from the CCL through empirical conversion relationships from a sample of captive juvenile loggerheads (n = 26) and green turtles (n = 12), for which SCL and CCL were measured (Figure A1, Appendix B). Descriptive statistics of sizes per species are presented in Table 1.

Theoretical growth curves were used for each species to estimate the potential growth of each released individual during the tracking duration and investigate the potential impact of carapace growth on tag attachments. All the growth curves follow von Bertalanffy equations with parameters specifically estimated for leatherback [38], loggerhead, Kemp’s ridley and green turtles [39].

### 2.4. Tag Specifications

The satellite transmitters used in our experiments were designed by Lotek Wireless Inc. (Havelock North, New Zealand), a manufacturer of fish and wildlife monitoring systems. The tags were designed to be extremely light in weight (e.g., ≈2–5 g), suitable for sea turtles weighing in the range of 100–150 g, in accordance with the <5% standard [13,21,22,23]. Novel technology, materials, batteries, and transmission strategies were utilized in order to achieve light weights and small “footprints” for the transmitters.

The microsatellite transmitter maximizes the benefits offered by the Argos-3 network. Argos-3 is a quadrature phase shift keyed (QPSK) modulation scheme that implements a Digital Video Broadcasting (DVB) type convolutional encoding. Convolutional encoding offers a theoretical 3 dB gain through the transmission channel, which in turn means that transmit power can be halved for the same amount of data throughput and error rates. The Argos-3 specification also allows a so-called PTT-ZE message, which is an ultra-short transmission containing transmitter ID information without sensor data. This further reduces the average power consumption of the transmitter. The above-mentioned benefits result in a reduced power requirement for the transmitter, thereby enabling the use of smaller batteries, which are often the largest and heaviest part of a satellite transmitter design.

In these experiments, 3 different models were used: the K4H 132A model, the K4H 130B model and the K4H 130B Dive model (Figure 2). Each model is constructed using an external housing machined from lightweight foam, which, when filled with the electronics and fully potted using epoxy, is capable of withstanding depths of 200 m. The K4H 130B and the K4H 130B Dive model each have solar harvesting capabilities, which may extend the life of the tags. Alternative tags built by Lotek using the same harvesting technology have achieved >2 years of deployment life in other applications [40]. The K4H 130B Dive model additionally offers a pressure sensor, allowing for individual dive records, daily summaries, and depth histogram data to be collected and compressed on-board the tag for eventual transmission through the Argos network. A whip antenna is used to transmit to the Argos satellites such that a transmission is attempted only when the tag detects that the antenna is clear of the water via a saltwater switch (or wet–dry switch).

All tags featured an accurate real-time clock. This allowed for the programming of intelligent scheduling of tag transmission during a time when satellites were predicted to be overhead. It also allowed for the implementation of “off-days” programming, i.e., days when the tag was scheduled not to transmit. This type of programming can act to further extend the expected lifetime of the tags.

Hereafter, the K4H 132A model will be referred to as non-solar (NS) tag and the K4H 130B and K4H 130B Dive models will be referred to as solar (S) tag.

### 2.5. Optimization of Transmission Schedules

Tag programming enables users to adjust the transmission schedule for the tracking period. On such miniaturized transmitters, it is fundamental to optimize tag transmissions in order to save battery and extend their lifetime. In this regard, optimized transmission schedules were determined to limit their transmissions to times when it was expected that satellites would be overhead. Depending on the expected dispersal area around the different release sites, we estimated the satellite coverage hours from Argos website data and fitted the transmission schedule to those coverage hours. To maximize transmissions prior to potential tag failure (e.g., tag attachment or operation failure), most of the non-solar tags were programmed to transmit more often during the first month of deployment.

### 2.6. Attachment Methods

For all species, the attachment site was washed with cotton gauze soaked in antimicrobial soap and chlorhexidine gluconate solution (0.25%). Any remaining loose scales or scute material was removed via light buffing with a terry cloth towel. The site was then rinsed with fresh water and dried with a clean towel. Tag attachment was species-specific from this point forward.

Two attachment methods were used for green turtles. With the first method, hereafter designated as Green 5200 (or CM5200), after cleaning, loose keratin was removed by gently sanding using 400 grit sandpaper, taking care to avoid abrasion of the underlying integument. The site was wiped clean with 70% isopropanol wipes and allowed to dry. A thin bed of 3M™ Marine Adhesive Sealant 5200 Fast Cure (3M, Saint-Paul, MN, USA, hereafter, 3M 5200 FC) was applied to the base of the tag and the tag attachment site, then the tag was attached to the turtle. An additional 3M 5200 FC was applied around the edges of the tag, avoiding the saltwater switches, to secure and shape the attachment. The adhesive surface was smoothed using an ice cube. When dry, a thin layer of black 3M™ 5200 Marine Adhesive Sealant (3M, Saint-Paul, MN, USA, hereafter, 3M 5200) was applied to the attachment for camouflage. The second method, hereafter designated as Green Epoxy (or CMEPO), was used with larger juveniles. After cleaning the carapace with soap and water, 80 grit sandpaper was used to lightly roughen the attachment site. Care was taken to avoid damaging the underlying integument. The site surface was wiped with fresh water and then thoroughly dried with a towel. The area was then cleaned with rubbing alcohol (70% isopropanol) and dried with a towel/allowed to air dry. A two-part epoxy (Superbond^TM^, Superclear, Saint Petersburg, FL, USA) was applied to the tag attachment site and the base of the tag and then the tag was attached to the turtle. Epoxy was smoothed around the tag, avoiding the saltwater switches.

Leatherback tag attachments varied slightly between the years. In year 1, the site was wiped clean with 70% isopropanol and allowed to dry. A neoprene (wetsuit material) pad was cut into strips and attached to the clean shell with Perma Rite #1 Plus (On Rite, Fort Lauderdale, FL, USA) cosmetic adhesive positioned immediately lateral to the mid-dorsal carapacial ridge to create a platform for the tag. The tag was attached to the neoprene with Perma Rite #1 Plus adhesive. The tag was anchored to the adjacent ridges and to the neoprene pads with a 5-0 PDS (Polydioxanone) suture. 3M 5200 FC was applied to cover the tag base, neoprene, and to form a fairing that was smoothed with an ice cube. Care was taken to avoid the saltwater switches. After the adhesive was no longer tacky, a thin layer of black 3M 5200 was applied to the attachment for camouflage. Hereafter, this method will be referred to as the Leatherback Year 1 (or DC1) method. In year 2, there were two changes to this attachment. PDS II 3-0 suture anchor was attached between the tag’s anterior corners through the neoprene and the two lateral ridges. There were no posterior suture anchors. Otherwise, the attachment was the same as in the previous year. Hereafter, this method will be referred to as the Leatherback Year 2 (or DC2) method. Two further changes were made in year 3. These included the elimination of the neoprene pad and substitution with an elastane fabricpad. PDS II 3-0 suture was used to form a continuous loop anchoring the tag to the ridges in 4 points and passing through the front and back channels in the tag housing. Hereafter, this method will be referred to as the Leatherback Year 3 (or DC3) method.

The attachment methods for small juvenile loggerhead and Kemp’s ridley’s followed a previously published method for 100–800 g neonates [15]. Briefly, after washing the carapace with an antimicrobial solution, loose scute material was sanded smooth with 320, then 400 or 600 grit sandpaper, taking care to not abrade the integument. The site was wiped clean with 70% isopropanol. The scute material was stabilized with a thin layer of manicure acrylic. A neoprene pad (wetsuit material) was cut into strips and attached to the acrylic layer with Perma Rite #1 Plus cosmetic adhesive to create a platform for the tag along or between carapacial spines or keels. Once dry, Aqueon™ (Aqueon, Franklin, WI, USA) nontoxic aquarium silicone adhesive was used to attach the tag and form a fairing around the tag base and sides. The adhesive sealant was not applied to saltwater switches. The adhesive was smoothed using an ice cube. Hereafter, this method will be referred to as the Loggerhead/Kemp’s Ridley (or CC/LK) method.

Tag attachments to passive drifters were similar except as noted here. Bucket drifters had four holes drilled into the buckets at the tag attachment site in the approximate shape and size of the tag. The fishing line was threaded through the drilled holes and through the tag’s anchor holes and tied as an initial attachment to the bucket. Then, the protocol for tag attachment to bucket drifters and manufactured drifters was the same: the bucket surface was sanded with 400 grit sandpaper and wiped clean with acetone and allowed to dry. A thin bed of 3M 5200 FC was applied to the base of the tag and the tag attachment site, then the tag was secured to the drifter. More 3M 5200 FC was applied to the tag and the attachment site to secure the attachment. The application of adhesive sealant was carefully avoided on the saltwater switches. Hereafter, these methods will be referred to as the Bucket Drifter (or BD) and the Manufactured Drifter (or MD) methods.

### 2.7. Release Methods

A total of 164 prototype microsatellite tags were equipped on sea turtles (n = 160) and drifters (n = 4), released off Florida, off Grand Cayman (Cayman Islands), within the Azores archipelago, and off Jekyll Island. All release events were conducted by universities and local organizations listed below in collaboration with Upwell for logistics and with FAU assisting with tag attachment methods.

#### 2.7.1. Florida

Since 2020, nine release events involving turtles reared at FAU have been organized by Upwell and completed within the Gulf Stream plume east of Florida (two in 2020, three in 2021 and four in 2022). These release events included juvenile leatherbacks (n = 54) (Figure 3a), juvenile loggerheads (n = 13) (Figure 3b), juvenile green turtles (n = 6) (Figure 3c), and drifters (n = 4) (Figure 3d). The turtles were released with non-solar (n = 54) and solar tags (n = 19), including some with dive sensors (n = 9). The drifters were fitted with non-solar tags. For each release event, possible release sites were determined between West Palm Beach (FL, USA) and West End (Grand Bahama, Bahamas) to ensure that the turtles were advected into the Gulf Stream to examine the influences of the current (velocity and direction) on initial dispersal orientation, swimming speed, and behavior in further studies. Microsatellite tag attachment was completed 12–36 h prior to release and then turtles were transported during the day by boat to the release locations and held in closed insulated coolers during transport to prevent dehydration and hyperthermia. The release of green turtles was conducted at night for opportunistic reasons (boat availability).

#### 2.7.2. The Cayman Islands

Three release events took place in the Cayman Islands in 2022 and were facilitated by the CTCEC, Upwell, and the University of Maryland. During these release events, juvenile green turtles (n = 50) (Figure 4) were equipped with non-solar (n = 20) and solar tags (n = 30). Forty tagged green turtles were released by boat off the coast of Grand Cayman (n = 40) in two release events. The first release events occurred 10 km north of Rum Point (n = 30 turtles) and the second occurred 10 km south of Spotts Beach (n = 10 turtles) [37]. The remaining 10 turtles were released within an artificial lagoon in Grand Cayman in order to observe tagged sea turtles interacting with their environment and other sea turtles.

#### 2.7.3. Azores

Since 2021, four release events have been performed within the Azores archipelago (one in 2021 and three in 2022), orchestrated by Upwell, the Institute of Marine Research (IMAR), Flying Sharks, and the Okeanos Institute of the University of the Azores. During these release events, wild-caught juvenile loggerheads (n = 30) (Figure 5) were equipped with non-solar (n = 20) or solar tags (n = 10). Tags were attached 12–36 h prior to release. Turtles were transported during the day by boat to the release locations and held in separate boxes protected from the sun to prevent dehydration and hyperthermia. The turtles were slowly acclimated to the seawater temperature before release. The release locations were selected based on a combination of factors, including prevailing weather conditions, tidal flows, sea state, and proximity by vessel to Horta, Faial. The first three release events were located in offshore waters to the north of the Faial-Pico channel, while the last one was located south of Pico Island.

#### 2.7.4. Jekyll Island

During 2022, Upwell and the Jekyll Island Authority’s GSTC organized a release event off Jekyll Island. During this release event, juvenile Kemp’s ridleys (n = 7) (Figure 6) equipped with non-solar tags were split into 2 groups. Some turtles (n = 4) were released by boat 10 miles offshore, east of Jekyll Island, and the others (n = 3) were released directly off the oceanic beach on Jekyll Island.

### 2.8. Tag Performance Metrics

The following diagnoses were computed from technical data directly relayed by the tags. Individual values were averaged per tag to compare the inter-individual variability of each feature and per group of tags, combining species and tag types. Eight groups were formed: non-solar tags equipped on Kemp’s ridleys (NS-LK), on green turtles (NS-CM), on loggerheads (NS-CC), on leatherbacks (NS-DC) and on drifters (NS-drifter), and solar tags equipped on green turtles (S-CM), on loggerheads (S-CC) and on leatherbacks (S-DC). To calculate the average of specific data described below, for each group of tags, all the measurements transmitted by the tags in that group have been averaged together. As a result, a tag operating longer than others will have a greater influence on the average. This choice was made to analyze the data during the “normal” operation of the device and to avoid biasing the average with devices that suffered early failures.

#### 2.8.1. Transmission Features

Analyzing the transmission features in satellite-tracking systems is crucial for understanding the resulting performances. Two transmission features were investigated in this study to characterize tag functioning: transmission current and transmission power. These key parameters, among others, play a pivotal role in directly impacting the transmission efficiency, battery life, and data quality of satellite tags.

Transmission current, expressed in milliamperes (mA), indicates the current drawn from the cell to produce the transmission pulse. Since this information is directly relayed by the tag, a transmission current value is regularly provided (every 5th transmission) and then averaged per group of tags, as indicated previously. Conversely, transmission power is not directly relayed by the tag but can be approximated with another relayed information, best level. The best level, expressed in decibels (dBm), represents the highest power level received by the satellite from the tag during one satellite pass. As indicated previously, a value of the best level is provided for each satellite pass and then averaged. Data from tags that have not relayed enough measurements (n ≤ 30) of each transmission current and best level were discarded.

#### 2.8.2. Transmission Performances

The ultimate goal of these experiments was to obtain trajectories that allow for detailed studies of juvenile sea turtle dispersal and their behavioral responses to oceanic conditions. To achieve this goal, state–space models (SSMs) are commonly used to process the trajectories, enabling regular sampling despite location errors and data gaps [42]. However, in order to reduce uncertainties, the raw trajectories need to be as accurate and regular as possible.

The accuracy of the transmissions was investigated from the relayed location error radius. The location error radius represents the radius of the circle having the same surface as the error ellipse calculated by the Argos system. This number has the advantage of embodying a common value to all locations (e.g., regardless of the angle or the width of the ellipse), thus allowing a global comparison of the accuracy of all the received locations. Finally, in the case of multiple received locations for the same tag on a given day, only the best location (i.e., with the lowest error radius) was kept in this diagnosis to quantify the accuracy of received locations. The regularity of the transmissions was assessed with the Transmission Regularity Ratio (TRR), defined as follows:(1)$$TRR=\frac{n_{ETD}}{n_{STD}}$$
with n_STD_, the number of scheduled transmission days (STDs), representing the number of days when transmissions are scheduled depending on optimized transmission schedules, and n_ETD_ the number of effective transmission days (ETD), representing the number of STDs when at least 1 consistent location (with error radius < 100 km) was received. This diagnosis is a good indicator of the transmission frequency, considering the programmed transmission schedules and showing that the higher the ratio, the fewer STDs are missed and, therefore, the fewer gaps are present in the trajectories. To compute this TRR, tags that have not transmitted long enough (number of STDs ≤ 10) were discarded. Finally, the lifetime of the tags was computed simply by calculating the time difference between the release of the tag and the last received transmission.

### 2.9. Tag Failure Investigations

#### 2.9.1. Software Issues

On non-solar tags attached to leatherbacks and deployed in 2020 (n = 17), a software issue was detected, leading to early tag failures. The problem was revealed through transmissions occurring outside of the programmed transmission schedule, which caused failures after only a few days after the release. All of the failures from these tags were directly attributed to a software problem. This problem was resolved in 2021 and, hence, tags deployed in 2021 and afterward were not affected by this issue.

#### 2.9.2. Battery Exhaustion

Non-solar tags have limited batteries, which deplete over time. Their voltage levels can be approximated via the battery voltage data, which are recurrently relayed by the tags. A significant drop in this voltage, down to a critical level (3.3 V according to the manufacturer specifications), is a clear sign of battery exhaustion. If this drop is not identified or the number of received data does not allow its identification (n < 10), the battery exhaustion can also be detected through the count of the number of sent transmissions, which is also occasionally relayed by the tags. Depending on the transmission current (mA), the battery is designed to send a defined number of transmissions. Thus, from the mean transmission current of the tag, and only for those relaying enough transmission current measurements (n ≥ 10), it is possible to calculate a limit number of transmissions above which it is reasonable to conclude that the battery is exhausted. If the data do not show that this limit has been exceeded or if the data do not allow this excess to be identified, then it cannot be concluded that the battery is exhausted.

Unlike non-solar tags, solar tags are equipped with a solar panel, allowing solar harvesting and battery extension. However, fouling of the solar panel can rapidly appear and alter the solar harvesting capacity of solar tags. When this occurs, the tag cannot recharge, and the battery will exhaust. In the case of solar tags, a significant drop in battery voltage just before the last transmission is sought first. If this drop is not identified or the number of data do not allow its identification (n < 10), it cannot be concluded that the battery is exhausted. The identification of an alteration of the solar harvesting is needed to complete this diagnosis and to conclude to a battery exhaustion. To do so, the difference between the minimum and the maximum values of instantaneous battery voltage from the previous day is computed to estimate how much solar energy the tag was harvesting. If this quantity is dramatically decreasing at the end of the records, solar harvesting capacity alteration can be concluded and, associated with the significant drop of the battery voltage, it can be concluded that the battery is exhausted.

Solar tags equipped with diving sensors have the same features as solar tags but do not relay data of instantaneous battery voltage. Instead, these tags prioritize the relay of dive data. Therefore, solar harvesting capacity alteration cannot be diagnosed, and battery exhaustion cannot be concluded.

#### 2.9.3. Tag Damages and Stranding

Juvenile sea turtles, such as the Kemp’s ridleys and the larger green turtles released in this study, were recruited to neritic waters, where they likely interacted with rocks, reefs, and sea bottom in shallow waters. Behavior such as scratching the carapace against substrate or structure can significantly damage the attached tag by breaking the antenna or the housing, or by directly removing the tag by damaging its attachment [25].

In the present study, no technical data were relayed to identify or inform on this kind of tag failure; instead, it was investigated by comparing the last position from each trajectory with the associated bathymetry from the General Bathymetric Chart of the Oceans (GEBCO) dataset [41]. It was assumed that if the last position from a trajectory was identified within a 10 km range from an area where the depth was 30 m or less, the tag was potentially damaged by interaction with the environment. We chose a radius of 10 km around the last received position to consider the potential movement of the animal during the day following this last transmission. In fact, by definition, when the last transmission was received, the tag was still working, and a new transmission window should occur in the next 24 h. If the tag was no longer transmitting during the next transmission window, it failed during this period. Also, the chosen critical depth (30 m) was set based on diving data (see Section 3.4). These data showed that even the smallest (<15 cm SCL) sea turtles of the current study can actually dive to this depth range.

Ocean conditions can also cause the stranding of a sea turtle or a drifter and in this case, the tag can also be damaged or buried, causing the transmissions to stop. In this study, it was assumed that if the last position from a trajectory was identified on land, the tracked individual was stranded and thus, its failure was associated with tag damage.

#### 2.9.4. Biofouling (Saltwater Switch Fail)

Tags can only transmit their data when they are out of the water, so they are equipped with saltwater switches to indicate if the tags are submerged (transmission not allowed) or dry (out of the water, transmission allowed). This reduces transmission inefficiencies by ensuring that tags only attempt transmission when conditions are favorable. In the marine environment, biofouling (accumulation of organic matter) can result in a malfunction with the saltwater switch, indicating that the tag is underwater when it is not and blocking the tag from transmitting data.

To investigate possible tag failure from biofouling, we compared individual tracking durations with the time of appearance of biofouling on tags reported in the literature. Recent studies show that biofouling was rarely the main cause for tag failures and appeared only after relatively long tracking durations [4,15,18,19,25,26]. In accordance with these studies and considering that the tags used in this study are miniaturized and probably more rapidly affected by biofouling than the other larger tags, the critical time for a saltwater switch failure was set to 150 days in this study. In other words, if the tracking duration was recorded to be longer than 150 days, we assumed that biofouling could have occurred and stopped the transmission ability of the tag.

#### 2.9.5. Tag Detachment Due to Turtle Growth

Growth rates of juvenile turtles are relatively rapid, especially for leatherbacks [38]. In this context, the tag attachment is severely tested since turtle carapaces are extending significantly, making the tag attachment very fragile and more likely to come off [26] under any kind of force (e.g., breaking waves, vigorous swimming or diving and surfacing movements).

Tag attachments are likely to fail when SCL growth reaches or exceeds 2 cm [43]. Hence, for each released individual, the critical time required for a 2 cm growth of the carapace from its release size was computed and compared to the tracking duration for that individual. If the tracking duration was between this critical time and 150 days (i.e., biofouling criteria), we assumed that the tag could have detached due to turtle growth.

Based on data from leatherbacks reared in the FAU laboratory, it takes 30 days for neonates in the range of sizes of the released individuals in our study to reach a 2 cm growth of SCL in water at 22–24 °C [44]. In the absence of in-lab data for the other species used within our study, the critical time to reach a 2 cm growth of SCL was computed with the release sizes and growth curves previously detailed.

### 2.10. Statistical Analyses

#### 2.10.1. Statistical Comparison

To determine whether or not certain technical factors have an impact on lifetimes, we performed Kolmogorov–Smirnov tests. Kolmogorov–Smirnov tests, applied to two samples of data, are statistical tests that determine if the two samples are drawn from the same distribution, based on the maximum difference between their empirical cumulative distribution functions [45]. For these tests, we used a 95% confidence interval, implying that if the resulting *p*-value is greater than 0.05, the defined null hypothesis (i.e., equality of distributions) cannot be rejected.

In this study, we investigated the potential impacts that tag types and attachment methods could have on tracking durations. We tested tracking durations for solar (n = 69) and non-solar (n = 95) tags, attachment on green turtles with the Green Epoxy (n = 25) and the Green 5200 (n = 31) methods, and the Leatherback Year 2 (n = 24) and the Leatherback Year 3 (n = 12) methods for leatherbacks turtles. Tags attached on leatherbacks with the Leatherback Year 1 method were discarded due to the aforementioned software issues.

#### 2.10.2. Tracking Duration Analysis

With tags for which the failure cause remained unidentified, a statistical analysis of lifetimes was performed in order to understand which failure type was likely responsible for the cessation of transmission. We performed a parametric statistical analysis, fitting reliability functions (number of working tags over time) with a Weibull distribution and investigating the associated failure rate functions. Weibull distribution is now a basis for lifetime studies thanks to its versatility to fit an important number of reliability functions [46] and being used in a large range of engineering fields such as the wind power industry [47] and material sciences [48,49].

From a sample of N systems associated with failure times, t_j_, we defined R(t), the reliability function as:(2)$$R\left(t\right)=1-\frac{card\left\{t_{j}\leq t\right\}}{N}$$
and the associated instantaneous failure rate:(3)$$\lambda \left(t\right)=-\frac{1}{R\left(t\right)}\cdot \frac{dR\left(t\right)}{dt}$$

It can be difficult to analyze the instantaneous failure rate since its signal, being derived from empirical data, is highly noisy and variable from one timestep to another. To simplify the analyses, the reliability function described in Equation (2) can be fitted with the reliability function of a Weibull distribution with the following form:(4)$$R_{fit}\left(t\right)=e^{-\frac{t^{\alpha }}{\beta^{\alpha }}}$$
with α, the scale parameter and β, the shape parameter, specific to each function to be fitted.

In this case, the reliability function is easy to derive and its continuity and absence of noise makes it easier to analyze. However, above the visual analysis of the failure rate function, the shape parameter, β, is decisive in the analysis and enables the identification of failure type. Depending on the value of β, the failure rate, λ, evolves differently and can be associated with one of the phases of the bathtub curve [50], typical for engineering systems [51]:If β < 1, the failure rate is decreasing, typical of the “burn-in” period. This kind of failure rate indicates an “early failure” and is typically associated with systems with manufacturing defects;If β = 1, the failure rate is constant, typical of the “useful life” period. This failure rate indicates a uniform failure probability in time and is typically associated with “random failures”;If β > 1, the failure rate, is increasing, typical of the “wear-out” period. It clearly indicates that the failure probability increases with time and is typically associated with “fatigue failures”.

## 3. Results

### 3.1. Transmission Features

The analysis of transmission power distribution in relation to transmission current (Figure 7a) revealed the presence of two distinct groups, effectively separating the non-solar tags from the solar tags. In general, non-solar tags demonstrated lower transmission currents and power levels compared to solar tags. This difference between non-solar and solar tags in terms of transmission power was mainly due to an antenna detuning issue. This issue, not observed in solar tags, consists in a difference in impedance between the tag and its antenna, related to the absence of a ground plane on such small devices and to the configuration of the tag (e.g., orientation of the board and battery, presence of solar panel or not, capacitive coupling of the circuitry to its surroundings). As a consequence, we observe a significant decrease in transmission current (approximately 1.4 times lower). Transmission current and power are closely linked, therefore, the decrease in current directly resulted in a reduction in transmission power (approximately 1.6 times weaker). Specifically, the average transmission current for non-solar tags was recorded at 78.14 mA, while the solar tags exhibited a higher average of 109.59 mA. Similarly, the average power level for non-solar tags was measured at −131.3 dBm, whereas the solar tags displayed a higher average of −129.3 dBm.

This important difference in transmission features was also observed among individuals of every species equipped, except for Kemp’s ridleys, which did not transmit enough data (Figure 7b). For loggerhead, leatherback, and green turtles, all equipped with either solar or non-solar tags, solar tags exhibited better transmission performances. Strikingly, green turtles showed the lowest and highest average power levels, depending on the tag type. Non-solar tags attached to green turtles measured an average power level of only −133.7 dBm. In contrast, solar tags reached an average power level of −119.6 dBm (Figure 7c).

### 3.2. Transmission Performances

#### 3.2.1. Location Error Radius

In total, 16,382 locations were received during all experiments. In selecting only the best daily position (n = 3439) for each tag, the average error radius was 2.71 km. Most of these locations exhibited a small error radius (Figure 8a), with 94.8% of these “best daily” locations having an error radius smaller than 7 km and 82.8% having an error radius smaller than 2 km.

Solar and non-solar tags were relatively similar in performance (Figure 8b), with error radius values of 2.39 km and 3.11 km, respectively. However, the interspecies variability was significantly different. Tags attached to drifters and loggerheads provided locations with mean error radii smaller than 2 km (0.83 and 1.66 km, respectively), while tags attached to leatherbacks and green turtles provided less accurate locations (mean error radii 3.20 and 5.61 km, respectively). Tags attached on Kemp’s ridleys showed the least accurate locations, with the highest mean error radius being greater than 10 km.

#### 3.2.2. Transmission Regularity Ratio

On average, the deployed microsatellite tags relayed locations on 76% of the scheduled transmission days. However, important differences were observed between solar and non-solar tags, with solar tags exhibiting a higher level of transmission regularity compared to non-solar tags (as indicated by the mean TRR, Figure 9). On average, solar tags relayed locations on 85.2% of the scheduled transmission days, while non-solar tags did so on only 62.9% of the scheduled transmission days.

Differences were also observed among different species and with non-solar tags. While the TRRs were all beyond 70% for species equipped with solar tags, variability was observed for species equipped with non-solar tags, with some having ratios near 50% or less and others having ratios around 90%. Thus, non-solar tags attached on Kemp’s ridleys, green turtles and loggerheads showed the lowest mean TRR (38.5%, 50.3% and 57.3%, respectively), whereas non-solar tags attached on leatherbacks and drifters showed a high mean TRR, around 90%.

Regardless of the tag type, a hierarchical pattern emerged among tags attached to green turtles, loggerheads, and leatherbacks. Tags attached to green turtles had the lowest TRR, whereas tags attached to leatherbacks had the highest TRR, and tags attached to loggerheads exhibited an intermediate TRR.

### 3.3. Tag Lifetime

#### 3.3.1. Tracking Duration

The overall average tracking duration of the tags deployed in these experiments was quite low, at only 27.2 days. The minimum tracking duration was 0.26 days, and the maximum was 192.3 days.

Notable variability emerged among the different species tracked regarding tracking durations and their reliability function (i.e., how the number of working tags changes with time) (Figure 10). On average, the shortest mean tracking durations were observed for tags attached to leatherbacks (mean: 9.1 days ± 7.1 SD). This was followed by tags attached to Kemp’s ridleys (mean: 13.0 days ± 2.9 SD) and tags attached to green turtles (mean: 17.6 days ± 15.6 SD). In contrast, tags attached on loggerheads and drifters showed dramatically longer tracking durations, (respectively, mean: 62.6 days ± 37.1 SD and mean: 50.6 days ± 10.8 SD). Of note, there was an important dichotomy observed for green turtles. The smallest individuals (n = 6), with sizes ranging from 9.82 to 10.81 cm, exhibited much longer tracking durations (mean: 54.5 days ± 13.1 SD) than the largest individuals (*n* = 50), with sizes ranging from 25.58 to 51.97 cm (mean: 13.2 days ± 8.3 SD).

Kolmogorov–Smirnov tests showed that tag types and tag attachment methods did not have a significant impact on tag lifetimes (*p*-values > 0.05). Comparisons of tracking durations of solar and non-solar tags (KS = 0.210, *p*-value = 0.068), tags attached on green turtles with the Green Epoxy and the Green 5200 methods (KS = 0.186, *p*-value = 0.679), and tags attached on leatherbacks with the Leatherback Year 2 and the Leatherback Year 3 (KS = 0.292, *p*-value = 0.434) methods did not differ.

#### 3.3.2. Identification of Failure Causes

Analysis of diagnostic results to identify causes of tag failure revealed that only 45% (n = 74) of the failures could be explained from our methodology (Figure 11), while 55% (n = 90) remained unexplained. The distribution of these failures varied among species. All failures on tags attached to Kemp’s ridleys (n = 7) and drifters (n = 4), as well as the majority (66%, n = 37) of failures on tags attached to green turtles, could be attributed to a possible cause of tag failure using our methodology. However, only 37% (n = 20) of failures on tags attached to leatherbacks and 14% (n = 6) of failures on tags attached to loggerheads could be attributed to specific causes from our diagnoses.

The primary cause of failure that could be determined in this study was damage to the tags, accounting for 65% (n = 48) of the attributed failures. This failure type affected all the tags attached to Kemp’s ridleys (n = 7) and the majority (63%, n = 35) of tags attached to green turtles. However, this failure was minor in the smallest green turtles (≈10 cm SCL, n = 6), affecting only one individual (17%), two loggerheads (5%) and three leatherback sea turtles (6%).

The second most common cause of failure was software issues leading to early tag failures. As previously noted, a software issue was detected in non-solar tags deployed on leatherbacks during 2020. All failures from these tags (n = 17) were attributed to the software problem, accounting for 10% of all the failures.

Further analysis revealed that only 5% (n = 8) of the failures were associated with battery exhaustion. While this was a minor cause of failure for all tags attached to sea turtles, it was a significant factor for tags attached to drifters, with 75% of failures attributed to battery exhaustion. Out of the four tags attached to drifters, three failed due to battery depletion, while the remaining tag was damaged when the drifter washed ashore on the coast of North Carolina.

We also observed an exception: one non-solar tag attached to a juvenile loggerhead, measuring approximately 12 cm SCL, exhibited the longest lifespan, being tracked for 192.3 days before ceasing data transmission. Interestingly, there were no indications of battery exhaustion or interactions that could have caused tag damage. This tracking duration is longer than the critical time, for this individual, allowing for a 2 cm growth in carapace length calculated as about 139 days. In addition, it is also longer than our criteria associated with the potential occurrence of biofouling on the wet–dry switch (150 days). Therefore, the suspected causes for this particular failure were either tag detachment due to the growth of the carapace or a saltwater switch failure caused by biofouling.

For the remaining unidentified tag failures (55% of all deployed tags), the statistical analysis using Weibull distributions (Figure 12) revealed that the three reliability curves (one per species) could be fitted with Weibull laws of different parameters (R² > 0.95). Overall, these results indicated different failure rates depending on the species. However, in all three cases the shape parameter, β, was superior to 1 (β_DC_ = 1.4; β_CM_ = 2.9; β_CC_ = 2.2), and indicated an increasing failure probability with time, typical of the “wear-out” period of an engineering system (Figure 12a). They clearly indicate that, regardless of the species, the main failure cause for these tags was associated with a fatigue process. Although the failure rates all increased over time, differences were notable in the way they increased with time. Indeed, for tags attached to leatherbacks, the failure rate rose very rapidly during the first week before failures increased a little more slowly over the following days. For tags attached to green turtles, it was almost the opposite. The failure rate increased at a slower rate during the first week, then rose sharply throughout the entire tracking period. Finally, for tags attached to loggerheads, the failure rate increased slightly and gently throughout the tracking period (Figure 12b).

### 3.4. Diving Data

Overall, out of the nine tags equipped with dive sensors and attached to young (SCL < 11 cm) green (n = 3), loggerhead (n = 3) and leatherback (n = 3) turtles, eight transmitted dive data that could be analyzed in the current study (Figure 13). One tag attached to a green turtle did not transmit data. While all the species spent most of their time (>60%) under the surface, species-specific differences were observed. The green and the loggerhead turtles spent, on average, all their time (100%) within the first 10 m below the surface, with occasional dives that were deeper and short in duration. Despite these deeper dives that could go down to 15 m for green turtles and over 50 m for loggerhead turtles, these two species reached an average maximum daily depth of 7.82 m and 7.04 m, respectively. In contrast, the leatherback turtles observed in the present study went slightly deeper, spending 96% of their time within the first 10 m below the surface and 4% of their time at depths of 10–20 m. Dives deeper than the first 10 m below the surface are recorded daily, making the average maximum daily depth deeper than for the other species, at 22.50 m. The deepest recorded dive for these leatherback turtles was at 33 m.

## 4. Discussion

### 4.1. Good Transmissions despite Unfavorable Features Due to Miniaturization

Our analyses overall revealed that the microsatellite tag prototypes transmit at very low power levels. These weak signals pose challenges for satellites to detect, as they are more susceptible to various sources of perturbations affecting Argos transmissions. These perturbations may arise from technical or environmental factors, such as spatially dependent interferences (ambient noise), alignment issues between the tag and the satellite (transmission geometry), adverse weather conditions (heavy rain, cloud cover), or physical obstructions (waves, mats of macroalgae, dense vegetation, mountains, buildings) between the tag and the satellite. This situation is particularly pronounced for non-solar tags, as they suffer from antenna detuning, leading to a significant decrease in transmission current and power. Consequently, non-solar tags are more sensitive to the aforementioned perturbations.

Despite these challenges, the tags performed satisfactorily, regularly providing accurate locations. Compared to typical Argos location errors (<250 m to ≅2 km), the location errors generated by our tags were relatively large but within an acceptable range given limitations such as low transmission current and power. Over 80% of the best daily locations fell within an error radius range of 2 km or less.

The overall transmission regularity was satisfactory, providing consistent locations on 3 scheduled transmission days out of 4. However, there was substantial variability observed between tag types and species. Solar tags had better performance in terms of transmission regularity compared to non-solar tags, and this trend was mirrored in the transmission power hierarchy between the two types. Similarly, non-solar tags attached to leatherbacks showed better performance compared to non-solar tags attached to loggerheads, and non-solar tags attached to loggerheads showed better performances compared to non-solar tags attached to green turtles. This trend was also mirrored in the transmission power hierarchy between the species.

The variability in transmission regularity could also have been influenced at a smaller scale by either the dispersal area or specific turtle behaviors. Individuals of some species, like larger green turtles and the Kemp’s ridleys we tagged, exhibited neritic behaviors such as potentially spending more time at depth and less time at the surface compared to epipelagic oceanic turtles, and remaining close to the coast where obstructions such as onshore high vegetations or reliefs may have hindered transmissions [52]. Additionally, external disturbances, such as ambient noise from other electronic devices or radio signals, might further degrade transmission quality and disrupt signal reception. These factors could contribute to slightly better transmission performance in certain, more remote regions. For instance, the environmental electromagnetic noise in Argos frequency bands might be higher along the densely populated US east coast than in the vicinity of the Azores archipelago. Considering this, it was not surprising that tags attached to loggerheads, which exhibit epipelagic oceanic behavior and mainly disperse from the Azores, outperformed tags attached to Kemp’s ridleys or green turtles, which tend to have neritic behaviors and disperse in the Western Atlantic. Similarly, the better performance of tags equipped on surface drifters was not surprising, as they transmit at relatively higher power levels compared to other non-solar tags. It could also be attributed to their passive surface drifting behavior and the tag being kept out of the water for longer periods compared to small turtles that actively swim, dive, and submerge, voluntarily or not.

Overall, the transmission power of the tags remained the main factor affecting transmission regularity, resulting in a significant number of lost transmissions for the low-powered tags. However, other factors like turtle behavior and geographical distribution could also have influenced the performance of the tags.

### 4.2. Abnormally Short Tracking Durations, Which Do Not Allow for Correct Trajectory Analyses

The tracking durations observed in our experiments varied greatly depending on the species tagged. This supports that the behavioral habits of each species highly influence the functioning of the tag and, combined with the different species-specific attachment methods, its ability to last over time.

Drifters are passive, unable to control their own direction and speed independent of larger physical influences such as ocean currents, and all associated failures with drifters could also be classified as “passive”. Indeed, one drifter washed ashore, likely damaging the tag, or burying it in the sand, and the other three drifters experienced battery exhaustion. The passive nature of drifters distinguishes them from sea turtles, which subject the tags to different challenges based on the specific behavioral habits of each turtle and species (e.g., interactions with the environment, habitat selection, swimming, and diving behaviors). For tags attached on sea turtles, battery exhaustion is not a major cause of failure (3%), as they primarily stop functioning before it occurs.

Tracking durations of loggerheads were relatively long (average 62.6 days ± 37.1 SD) in comparison with other species in this study, and on the same order of magnitude as those observed for turtles of similar sizes (mean: 67.2 days ± 45.7 SD) in previously published studies [18,28]. While our diagnostics may not account for the majority of failures occurring on tags attached to loggerheads, the failures they do explain are primarily linked to passive failures, such as battery exhaustion, potential biofouling, or tag detachment due to turtle growth, which are more likely to occur during extended tracking periods such as observed in our study.

In contrast, tracking durations of tags attached on Kemp’s ridleys, green turtles, and leatherbacks were relatively short and well below the expected results. Although equipped with different types of transmitters (i.e., microsatellite tags), tracking durations observed in our study were quite short (<20 days) compared to tracking durations of similarly sized Kemp’s ridleys (mean: 114.4 days ± 91.7 SD) [27,29] and smaller green turtles (mean: 65.9 days ± 30.6 SD) [19]. This rapid decrease in the number of functioning tags attached to Kemp’s ridleys and green turtles could be explained by the limited capacity of our miniaturized tags and their attachment to withstand neritic “carapacial cleaning” and possibly biting of tags by conspecifics. In the current study, all tags attached to Kemp’s ridleys and the majority of tags attached to green turtles experienced failures associated with tag damage in shallow waters. This observation aligns with expectations considering that neritic Kemp’s ridleys and green turtles could exhibit significant environmental interactions, scratching their carapace against rocks, reefs, or the seafloor, and thereby exposing tags to significant damage to antennas, housings, or attachments [25]. Observations of other, non-released juvenile green turtles in the artificial lagoon of the CTCEC support this hypothesis, with a large number of interactions recorded between green turtles and rocks, as well as with other individuals, resulting in damaged tag antennas (Figure 14a), and damaged tag housings [53]. One of the juvenile green turtles released to the wild, offshore of Grand Cayman, had a tag that stopped transmitting but the turtle was found in good health about 10 days after the last transmission, with the upper part (tag upper housing and antenna) missing (Figure 14b). These observations confirm that the tagged juvenile turtles that reside in neritic rather than pelagic habitats can cause substantial damage to the tags, which could lead to early tag failures. The smallest tracked green turtles (≈10 cm) used epipelagic habitats and their tracking durations were dramatically longer and on the same order of magnitude as those observed for turtles of similar sizes [19]. These turtles are less likely to locate hard structures that can dislodge or damage the tags and likely can generate less force should an at-sea structure be available. This finding corroborates our observation that the microsatellite tags are challenged by neritic interactions when on larger turtles but can tolerate the movements and behaviors of smaller individuals.

Tracking the duration of leatherbacks was the shortest and the most surprising. This is the first satellite tracking of very early stage (~3 months) leatherback turtles of this size class (~100–150 g). We hoped for longer tracking durations, at least equivalent to those of the smaller loggerheads and green turtles from other published studies [18,19,28]. On the contrary, the average tracking time for small leatherbacks did not exceed 10 days, and the longest trajectory obtained was barely a month. While the short tracking durations of Kemp’s ridleys and green turtles could mainly be explained by the behavior of these species in shallow waters, leatherbacks exhibit different behaviors, being more pelagic and thus exposed to limited opportunities to interact with hard substrate [54]. The at-sea behavior of small leatherbacks, particularly vertical movements and time spent submerged, also differs from that of loggerheads, Kemp’s ridleys, and green turtles, therefore making it difficult to diagnose whether stresses on the tag or the attachment accounted for such early failures [55].

Although natural predation may have certainly played a role in the loss of some deployed tags, potentially impacting the turtle’s escape capability, it would be surprising if it were the main cause. In our study, loggerhead turtles were tracked for substantially longer durations than leatherback turtles of similar size despite being released in the same area, off Florida. Tags deployed on leatherbacks rarely lasted more than two weeks, and those deployed on green turtles rarely lasted more than a month. This rate of loss is far too high to be mainly due to predation and is inconsistent with the estimated sustainable survival rate for a given population, which would be around 25% after the first year at sea [56].

From a statistical perspective, our results demonstrate that predation has only a minor role compared to other causes of tag failure. In fact, the probability of being hunted and consumed by a predator is highest for the youngest and smallest individuals and decreases over time as the turtle grows (the bigger, the safer!) and its swimming and escape abilities improve. If predation played a major role in the loss of our tags, the failure rates would have decreased over time. For the important remaining fraction of tags for which the failures are still unexplained (65%), the statistical analysis demonstrated that regardless of the species, the failure rate increases with time, clearly indicating a tag fatigue process as a primary cause.

Overall, the main type of failure affecting the tags deployed in our experiments was a fatigue mechanism, regardless of the species. For some tags, failure rates developed in different ways depending on the species, showing that the cause of fatigue had variability, and could be dependent on behavioral habits specific to each species. This, however, leaves the question of what specific behavioral differences among leatherbacks, loggerheads, and green turtles could have caused these variable failure rates.

One possibility is dive behavior, as it is known that juvenile leatherbacks are inherently good divers [57,58]. Moreover, the diving data obtained from young juvenile loggerheads, leatherbacks and green turtles during this study show that, at a similar size, young leatherbacks dive much deeper compared to green turtles or loggerheads. Although loggerhead (and potentially green) turtles are capable of occasional relatively deep dives (>30 m) at small sizes (≈10 cm), they spend almost all their time in the first 10 m of the ocean. In contrast, leatherback turtles spend much more time deeper and regularly dive deeper than 20 m. Although the deepest recorded dive in the present study was performed by a loggerhead turtle, recent observations suggest that early-stage leatherbacks are capable of much deeper dives than observed within this dataset [59].

This marked difference in diving behavior, along with observed variability in tag function and duration between species, leads us to hypothesize that the deep and repetitive dives of leatherback turtles may be a cause of early failures. Dives can potentially degrade the housing on the tags due to increased pressures, leading to water infiltration and permanent tag damage. However, tags are exposed to a triangle pressure profile during their design phase, being subjected to five cycles of rated pressure (i.e., 20 bars—equivalent to a depth of 200 m) and 1000 cycles of half-rated pressure (i.e., 10 bars—equivalent to a depth of 100 m) at an applied rate of 4 m/s. Additionally, tags experience 24 h of exposure to a static pressure 1.5 times the rated pressure (i.e., 30 bars—equivalent to a depth of 300 m) [60]. These rigorous pressure chamber tests, conducted by the manufacturer on all tags prior to deployment, would likely suggest that the pressure is not the primary factor causing early failures. A possible hypothesis about a relationship between turtle dive behavior and tag failures could be that the drag associated with alternating positive and negative diving speeds experienced by a turtle on its dive ascents and descents could have a fatiguing effect on the antenna integrity, potentially leading to breakage and therefore to tag failure. We also observed corrosion on the printed circuit board (PCB) of returned tags, initially intended to be deployed on birds. This corrosion was mainly due to moisture being trapped in the tag housing, but marine tags should not suffer from this effect, being solidly encased in epoxy and hypothetically impenetrable to moisture. In response, a new cleaning process has been implemented during tag manufacture and further results are needed to study its impact on tracking durations. It is also possible that early tag failures could be attributed to faulty attachment methodologies. We cannot definitively rule out this hypothesis, but it seems highly unlikely, since most tags attached to leatherbacks ceased transmitting before the predicted attachment component failures. Indeed, tag attachment methods used for leatherbacks employ “hybrid” attachment techniques (i.e., suture anchors, various adhesives) and each has a different failure time. In-lab tests showed that the cosmetic adhesive lasts more than 4 months when attaching the tag to spandex and lasts until the scales are shed during turtle growth (about 30 days here). Further, the combination of suture anchors that supplement the cosmetic adhesive should last at least 3 weeks [61] based on lab observation of the integrity of the components.

While the deeper diving behavior of juvenile leatherbacks might result in malfunction or strain for deployed microsatellite tags, the shallower dives of juvenile loggerhead turtles should not have had these same issues. Instead, the fatigue process for loggerhead tags demonstrated a much more gradual increase over time. As the smallest green turtles in our study exhibited diving behavior and tracking durations similar to those of the loggerheads, they are likely to have a similar fatigue process. However, older green turtles have more pronounced and deeper dive behavior [62,63], potentially resulting in similar issues as the leatherbacks with the integrity of tags or their attachments. Additionally, the neritic, shallow spatial use of the larger green turtles of Grand Cayman in our study [37] could have resulted in some initial damage to tags that is then compounded when they venture into deeper oceanic areas where the tag fatigue process could speed up, leading to an increased failure rate over time.

## 5. Conclusions

We investigated the performances of prototypes of microsatellite tags for tracking neonate to juvenile sea turtles, with the aims of (i) assessing the performance of novel microsatellite tags and (ii) gaining insights into turtle behavior during this critical life history period. Our research confirmed that tag miniaturization for these small and highly mobile denizens of the marine realm poses real and ongoing technological challenges, imposing strong design constraints on the tracking devices. Tags for small sea turtles must be compact and lightweight to minimize any potential impacts on mobility and behavior. This constraint limits the space available for sensors, batteries, and other components, requiring careful selection and optimization of each element to ensure proper functionality and longevity. Miniaturized tags must be robust enough to endure the harsh and corrosive marine environment and to withstand the physical stresses imposed by the vigorous behaviors (e.g., diving, swimming, benthic foraging, carapace scraping, nipping) of young sea turtles, some of which subject the tags to continuous wear. The miniaturization process necessitates the use of smaller batteries to power the tags, which impacts both operational lifespans, as smaller batteries have reduced energy storage capacity, and the transmission power. Striking the right balance between battery, transmission power, robustness, size, and weight is a delicate task that requires extensive testing and optimization.

Our research findings revealed that the deployed microsatellite tag prototypes operate with very low transmission power. These low transmission powers induce weak signals, making them more susceptible to various sources of perturbations that can affect Argos transmissions. The impact of these unfavorable transmission features is particularly noteworthy on non-solar tags, which face an antenna detuning issue leading to a significant decrease in transmission current and power and rendering them more sensitive to the aforementioned perturbations. Despite these challenging transmission features, our experiments have yielded encouraging results in terms of transmission regularity and quality. The deployed microsatellite tags demonstrated commendable transmission regularity, with relatively few missing transmission windows, especially for solar tags. This is a crucial factor for obtaining accurate and consistent tracking data. Furthermore, the location error radius, which indicates the accuracy of the obtained locations, has proven to be low, underscoring the ability of our tags to provide reliable location information despite their weak signals.

One of the substantial findings of our research is that the main weakness of our microsatellite tags is not their battery capacity (as is documented for tags on adult sea turtles [4]), as they stop transmitting before battery exhaustion occurs. Instead, their primary limitation lies in their lack of robustness. The tags are too fragile and are unable to withstand the physical constraints encountered by individuals occupying neritic habitats, especially when deployed on large (>20 cm) juvenile green turtles and Kemp’s ridleys, which frequently interact with the sea bottom, rocks, and reefs. Juvenile sea turtles, especially those exhibiting neritic behavior, engage in various activities that expose the tags to significant wear and potential damage. Green turtles and Kemp’s ridleys, in particular, are known to scratch their carapaces against rocks, reefs, and the seafloor, which subject the tags to intense forces and abrasions, resulting in damage to the tag’s antenna, housing, or its attachment. Consequently, these physical threats to the tags on neritic turtles lead to a rapid decrease in the number of functioning tags attached to green turtles and Kemp’s ridleys.

The lack of robustness in the microsatellite tags presents a significant challenge for applications for actively diving oceanic turtles, such as neonate leatherbacks. The insights gained from our diving data have been groundbreaking, providing the first reported longer-term (greater than 3 days) observations of horizontal and vertical movements of small juvenile leatherbacks, loggerheads, and green turtles. Our diving data clearly illustrate the stark differences in diving behaviors among these sea turtle species. Small juvenile leatherbacks exhibited a distinctive pattern of more frequent and deeper dives compared to loggerheads and green turtles of similar sizes. These intense and repetitive dives likely subject the tags to high levels of stress and fatigue, potentially compromising the integrity of the tag or its attachment to the turtle. The vertical movements of small leatherbacks during these deep dives continuously subject the tags to pressure extremes that likely contributes to tag fatigue, resulting in water infiltration and degradation of the tag’s housing and antenna. The lack of robustness in our microsatellite tags is not only an issue for neritic individuals. It also poses significant challenges for diving oceanic sea turtles, particularly very young leatherbacks.

Our study underscores the need for a new phase of design and testing to address and overcome the existing constraints on the development of microsatellite tags. We have identified low transmission power and lack of robustness against neritic and diving behaviors as key challenges, and overcoming these limitations will require further innovation and technological advances. In order to improve the performance and durability of the tags, comprehensive design modifications and rigorous testing remain requirements. The current tags are well suited for continued research on small pelagic stage loggerheads and green turtles, as these species exhibited the longest tracking durations, consistent with previously published results [18,19,28]. These tags hold tremendous promise for revealing the vertical habitat use of very early-life stage leatherback turtles but they could be greatly enhanced through the innovations described here. Our overall aim is to optimize data collection efforts by concentrating on proven successes with the miniature tags and to continue to gather valuable insights into behaviors and movements during early sea turtle life history. As we embark on a new phase of design and testing, we must continue to approach the challenges with a multidisciplinary approach, drawing on expertise from biologists, engineers, ocean physicists, and conservationists. Collaborative efforts will be crucial to refine and enhance the tracking technology, ensuring its effectiveness in diverse marine environments and sea turtle species.

Despite challenges and limitations in data capture, retrieval, and analysis, our research efforts have achieved significant milestones in understanding the movements and behaviors of various sea turtle species. We successfully acquired the first reported longer-term horizontal and vertical tracking data leatherbacks at 70–100 day-old and early-stage captive-reared (Florida) and wild-caught (Azores) loggerheads. These groundbreaking results have addressed knowledge gaps surrounding the life histories of different sea turtle populations, particularly shedding light on the enigmatic “Lost Years” period in sea turtle life history.

Our learnings throughout the research design and implementation process have laid the foundation for refining satellite tracking efforts not only on early-stage sea turtles but also juveniles of other marine species, particularly those exhibiting complex life histories and ontogenetic changes. The insights gained from our research on microsatellite tags could have broader implications for the bio-logging community, extending beyond sea turtles to encompass a wide range of marine species. The challenges we have addressed, such as miniaturization, transmission power, and robustness of the tag and its attachment, are shared across various taxa inhabiting marine environments, such as fishes, marine mammals or even seabirds [13,14].

We have benefited from a unique interdisciplinary collaboration involving many scientific experts, including sea turtle biologists, oceanographers, physicists, electrical engineers, software engineers, and tag designers. Our multidisciplinary approach and collaborative efforts exemplify the importance of cross-disciplinary cooperation in advancing tracking technology for the benefit of marine conservation and ecological research.

## Figures and Tables

**Figure 1 animals-14-00903-f001:**
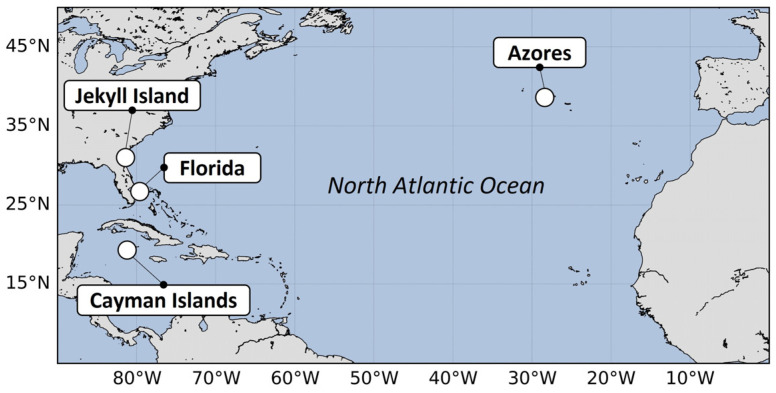
Map of the North Atlantic Ocean summarizing all the release sites used since 2020 to deploy the microsatellite tags (n = 164) analyzed in the present study.

**Figure 2 animals-14-00903-f002:**
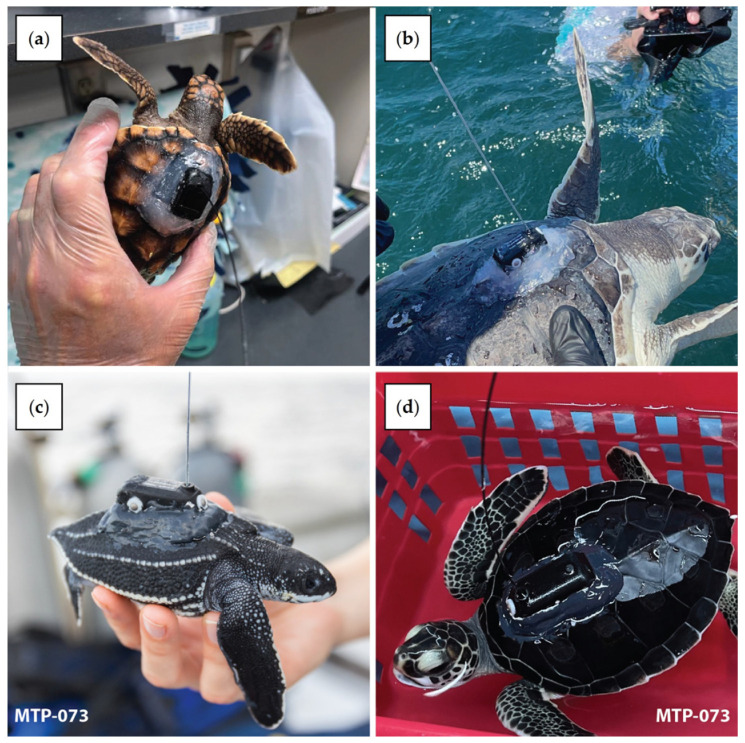
Juvenile loggerhead (**a**), Kemp’s ridley (**b**), leatherback (**c**) and green (**d**) turtles equipped with prototypes of solar (**a**) and non-solar (**b**–**d**) microsatellite tags. Images provided by Emily Turla (**a**,**d**), Jekyll Island Authority (**b**) and Jay Paredes (**c**).

**Figure 3 animals-14-00903-f003:**
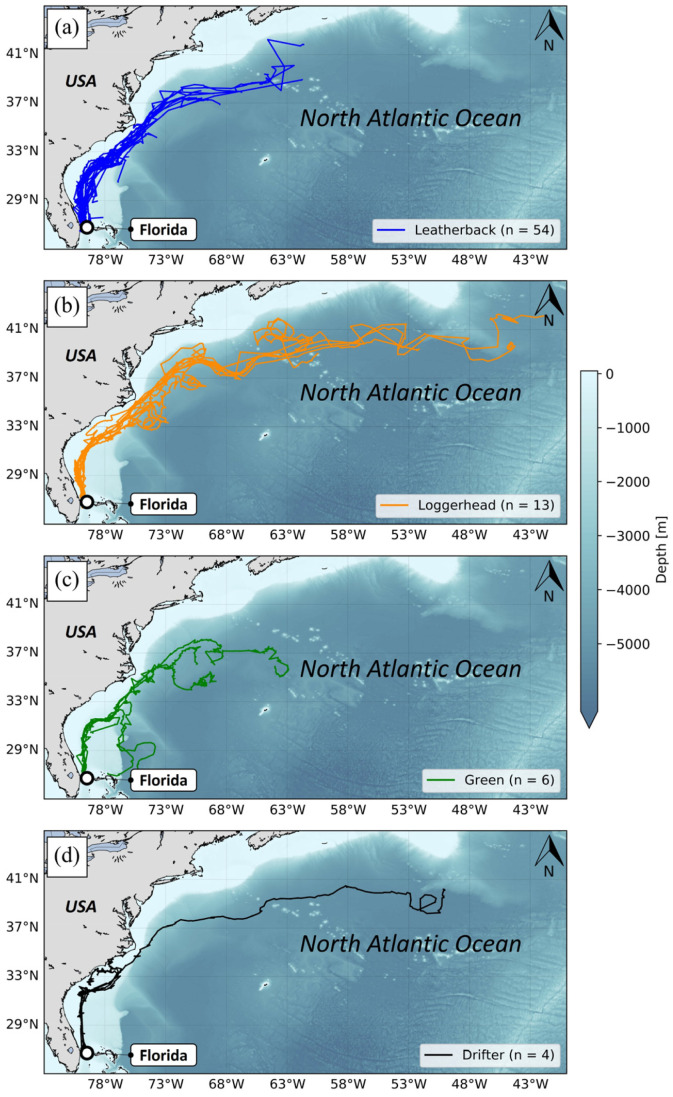
Trajectories of (**a**) leatherbacks, (**b**) loggerheads, (**c**) green turtles and (**d**) drifters obtained from Florida, superimposed with the bathymetry from General Bathymetric Chart of the Oceans (GEBCO) dataset [41].

**Figure 4 animals-14-00903-f004:**
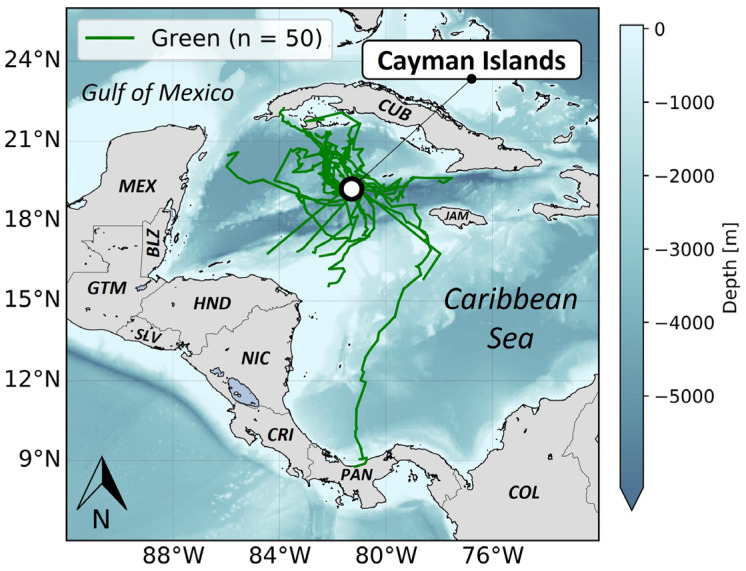
Trajectories of green turtles obtained from Cayman Islands, superimposed with the bathymetry from General Bathymetric Chart of the Oceans (GEBCO) dataset [41]. Countries around the area are indicated: Mexico (MEX), Belize (BLZ), Guatemala (GTM), Honduras (HND), El Salvador (SLV), Nicaragua (NIC), Costa Rica (CRI), Panama (PAN), Colombia (COL), Cuba (CUB), and Jamaica (JAM).

**Figure 5 animals-14-00903-f005:**
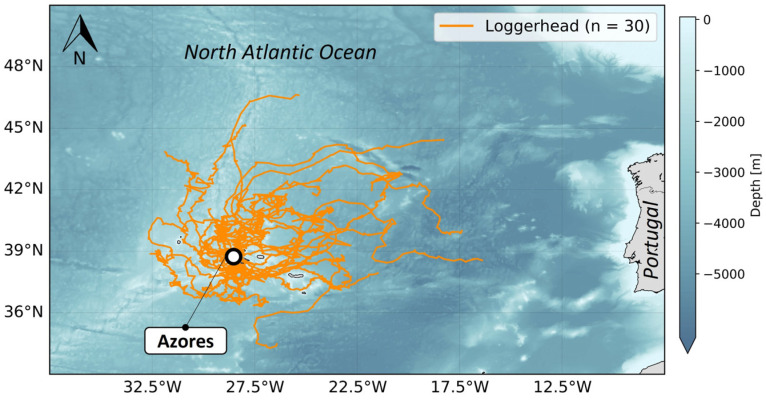
Trajectories of loggerheads obtained from the Azores, superimposed with the bathymetry from General Bathymetric Chart of the Oceans (GEBCO) dataset [41].

**Figure 6 animals-14-00903-f006:**
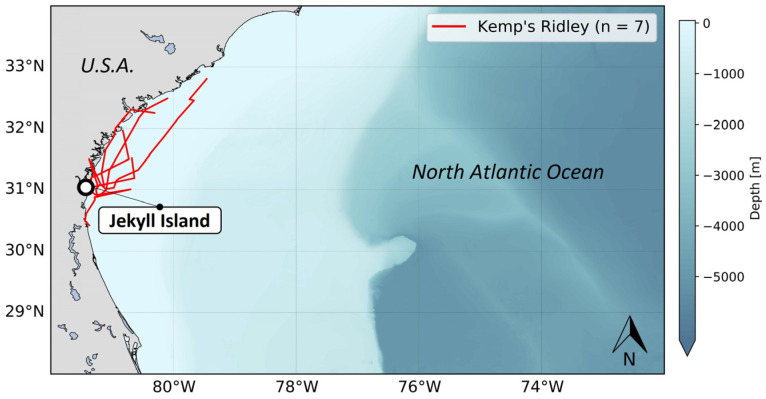
Trajectories of Kemp’s ridley turtles obtained from Jekyll Island, superimposed with the bathymetry from General Bathymetric Chart of the Oceans (GEBCO) dataset [41].

**Figure 7 animals-14-00903-f007:**
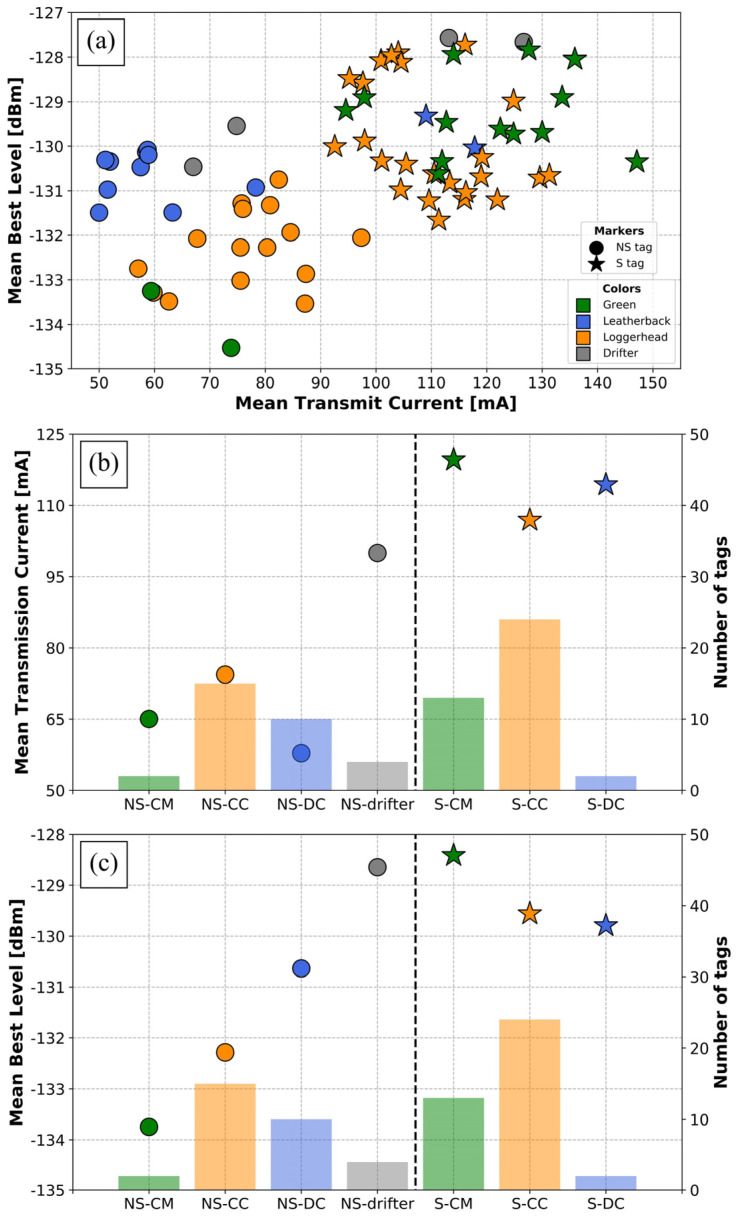
Combined analysis of the mean transmission current and the mean best level from tags with at least 30 measurements of each (n = 70). Panel (**a**) shows the mean best level as a function of the mean transmission current. On panels (**b**,**c**), tags are gathered by groups combining species (CM: green turtles; CC: loggerheads; DC: leatherbacks; and Drifters) with tag types (NS: non-solar; S: solar). They show, respectively, the mean transmission current (**b**) and the mean best level (**c**) per group of tags. The columns indicate the number of tags included in the average computation and the thin vertical black dashed line separates NS from S tags. For all diagnoses, markers indicate the type of tag and colors indicate the species.

**Figure 8 animals-14-00903-f008:**
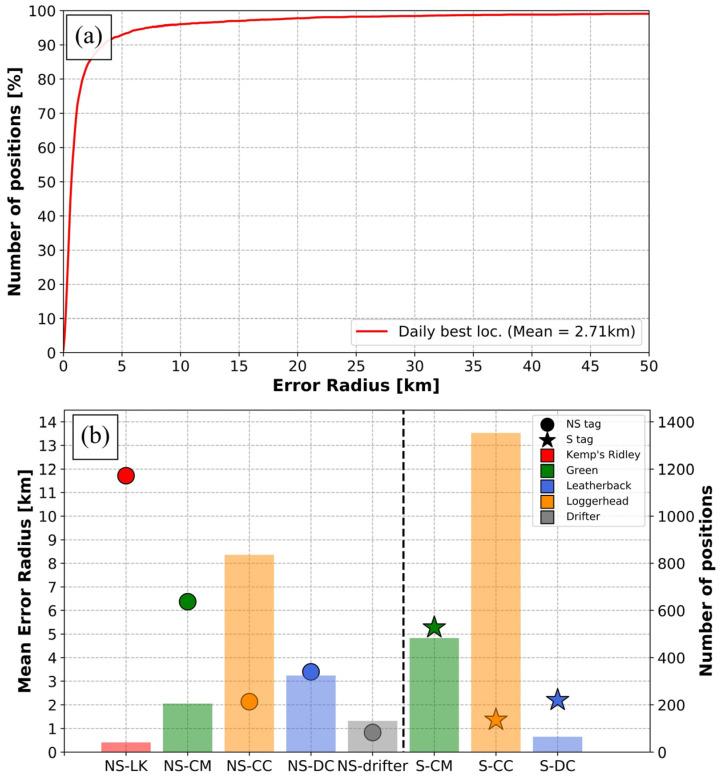
(**a**) Cumulative number of best daily positions (n = 3439) as a function of their error radius and (**b**) the mean error radius of these best daily positions gathered by groups combining species (LK: Kemp’s ridleys; CM: green turtles; CC: loggerheads; DC: leatherbacks; and Drifters) and tag types (NS: non-solar; S: solar). Columns indicate the number of positions included in the average computation, the thin vertical black dashed line separates NS from S tags, markers indicate the type of tag and colors indicate the species.

**Figure 9 animals-14-00903-f009:**
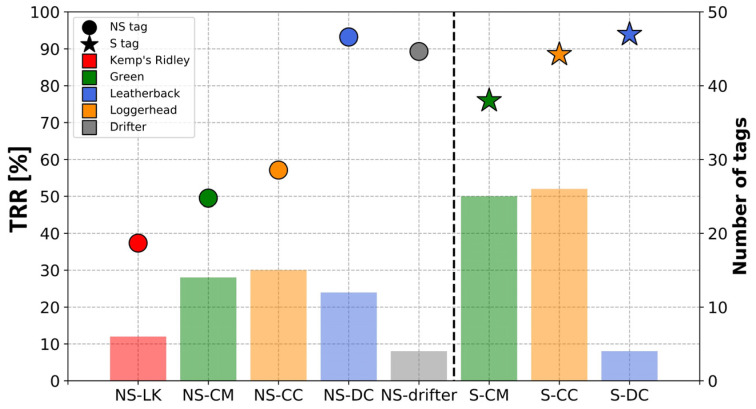
Mean transmission regularity ratio (TRR) per group of tags (n = 106) combining species (LK: Kemp’s ridleys; CM: green turtles; CC: loggerheads; DC: leatherbacks; and Drifters) and tag types (NS: non-solar; S: solar). Markers indicate the type of tag, colors indicate the species, columns indicate the number of tags included in the average computation and the thin vertical black dashed line separates NS from S tags.

**Figure 10 animals-14-00903-f010:**
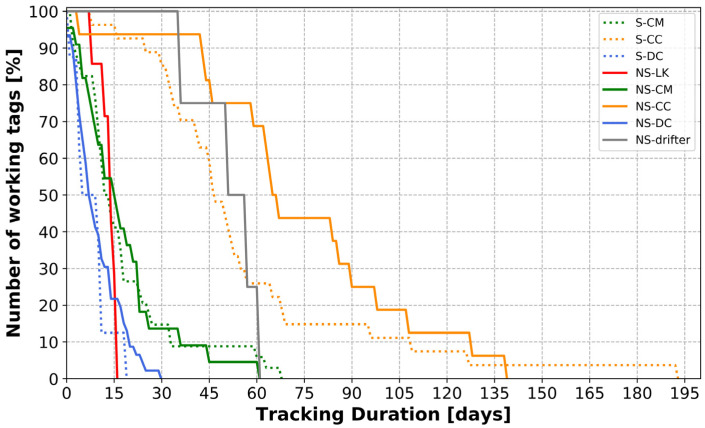
Attrition curves depending on group of tags combining species (LK: Kemp’s ridleys; CM: green turtles; CC: loggerheads; DC: leatherbacks; and Drifters) indicated by colors, and tag types (NS: non-solar; S: solar) indicated by line styles (solid or dashed lines).

**Figure 11 animals-14-00903-f011:**
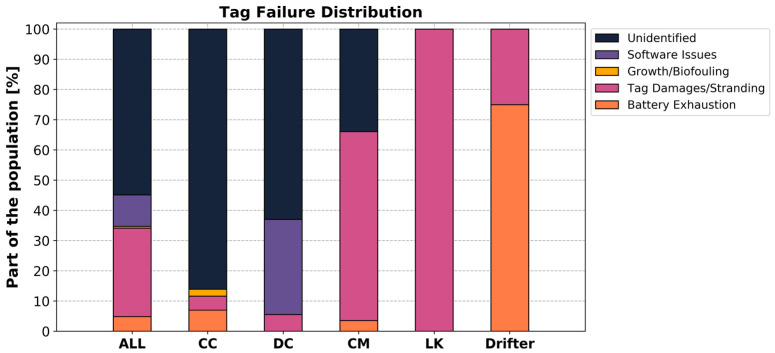
Histogram showing tag failure distributions depending on the equipped species (ALL: all four species; CC: loggerheads; DC: leatherbacks; CM: green turtles; LK: Kemp’s ridleys; and Drifters).

**Figure 12 animals-14-00903-f012:**
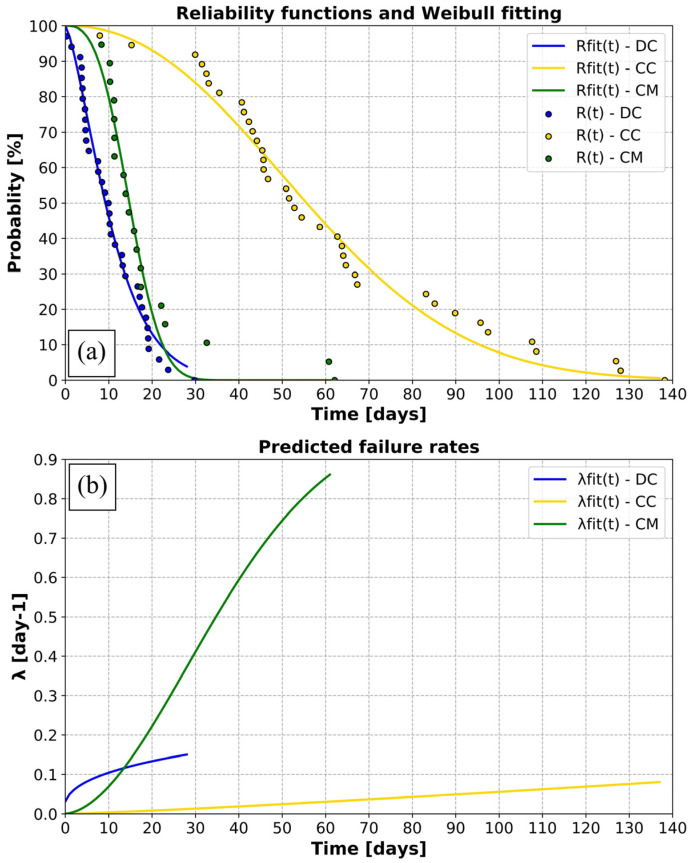
Reliability functions (**a**) and associated predicted failure rates (**b**) per species only, including tags for which the failure cause remains unidentified. On panel (**a**), reliability curves from data, R, are plotted as a scatter plot with one marker on each failure event. The fitted reliability curves from the Weibull approximation, Rfit, are plotted as solid lines. On panel (**b**), predicted failure rates functions, calculated from the Weibull approximation, are plotted as solid lines and until the last failure event.

**Figure 13 animals-14-00903-f013:**
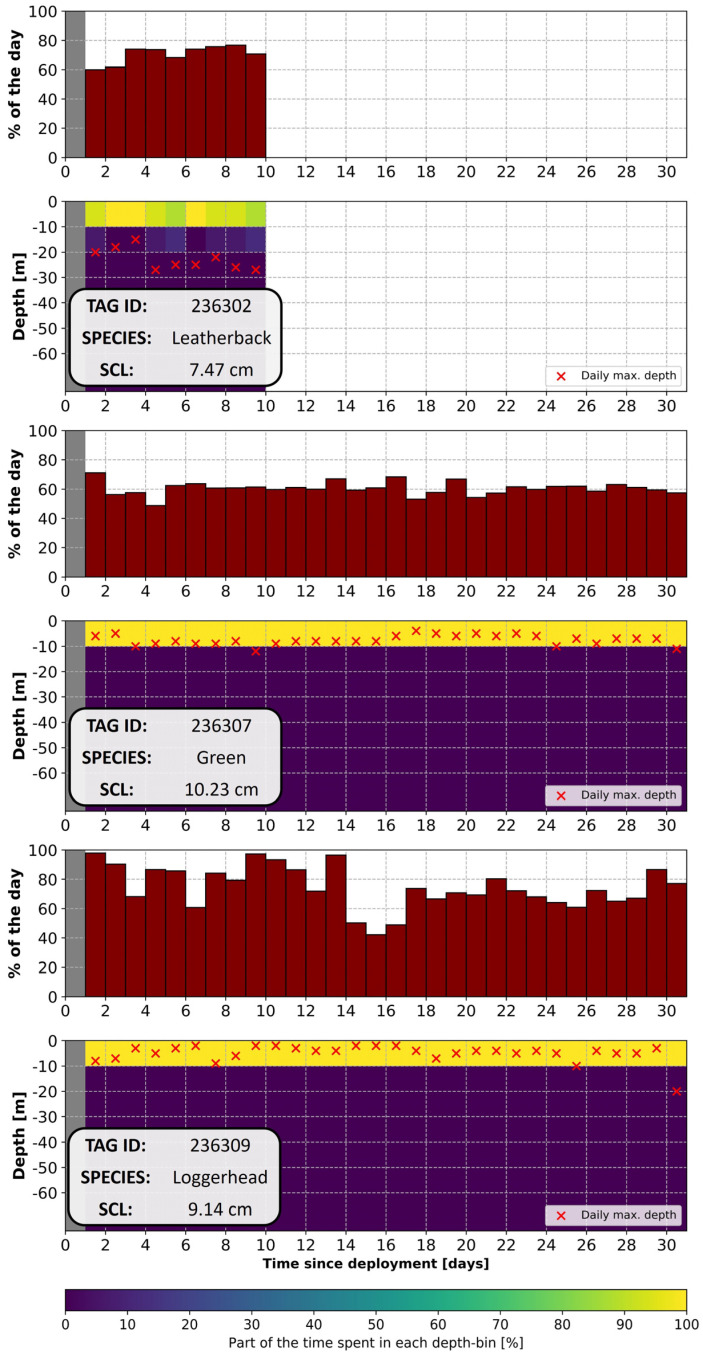
Diving data from three sea turtles tagged with solar microsatellite tags, equipped with dive sensors. Each figure is labeled with the tag ID, the equipped species, and the straight carapace length (SCL) of the individual. On each, two panels indicate the daily fraction of the day spent underwater (**top**) and the daily dive histogram (**bottom**), with the fraction of the time spent underwater spent in each 10 m depth bin. Red crosses indicate the daily maximum reached depth.

**Figure 14 animals-14-00903-f014:**
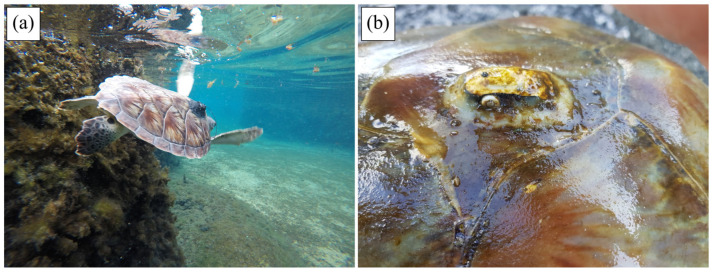
Observations of tag damages on green turtles from Grand Cayman. Picture (**a**) shows the remaining part of the tag with a missing superior part (housing + antenna) from the turtle found in the inlet channel a few days after the transmission ceased. Picture (**b**) shows a swimming turtle in the artificial lagoon of Grand Cayman with a broken antenna. The photographs were provided by Francesca Casella.

**Table 1 animals-14-00903-t001:** Descriptive statistics of straight carapace length of telemetered sea turtles depending on the species.

Species	Straight Carapace Length [cm]			
	Mean	Std.	Min.	Max.
Leatherback	8.78	0.59	7.14	9.65
Loggerhead	15.57	4.38	9.14	30.90
Green	32.68	11.21	9.82	51.97
Kemp’s ridley	27.79	2.64	24.90	32.50

## Data Availability

Restrictions apply to the availability of these data. Data were obtained from Upwell Turtles and are available (by request to the corresponding author and George L. Shillinger, george@upwell.org) with the permission of Upwell Turtles.

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
