# Peer review of "Novel Microsatellite Tags Hold Promise for Illuminating the Lost Years in Four Sea Turtle Species"

_animals, 2024, doi:10.3390/ani14060903_

Round 1

Reviewer 1 Report

Comments and Suggestions for Authors

See attached.

Reviewer 2 Report

Comments and Suggestions for Authors

The manuscript entitled "Novel microsatellite tags hold promise for Illuminating the Lost Years in four sea turtle species" provides an assessment of this developing technology for tracking smaller size classes of sea turtles. The manuscript provides a rather detailed methodological account of satellite tracking technology and could benefit from condensing the technical descriptions in some sections. The attached file provides detailed comments and suggestions with a few general comments below.

There is ambiguity in referring to the satellite transmitters as "tags" given that passive forms of tracking sea turtles are commonly referred to as such (i.e., external flipper tags and internal PIT tags). Suggest referring to the active tracking devices as "transmitters" or "PTTs" (as in line 97).

The sizes (carapace lengths) for Kemp's ridley turtles presented in Table A4 and summarized in Table 1 represent that commonly seen in the neritic stage (i.e., greater than 20-25 cm), not the oceanic stage. Furthermore, the initial stranding location suggests these turtles had already recruited to a nearshore foraging ground (Cape Cod Bay). As such, this sample does not necessarily represent the "lost years" for this species as indicated in the title and lines 115-117. The same applies to some of the larger green turtles as was indicated in the manuscript. These discrepancies should probably be addressed in the stated goals/objectives of the study (i.e., includes post-pelagic recruitment to the neritic zone for some species) and included in the appropriate subsections of section 2.7 Release methods or section 2.3 Turtle sizes and growth curves. Include content and references on recruitment to neritic habitats in the Introduction.

Comments on the Quality of English Language

Some improvements are needed in grammar and sentence structure throughout the manuscript.
